# MULTI-AGENT PATH FINDING VIA DECISION TRANS-FORMER AND LLM COLLABORATION

## ABSTRACT

Multi-Agent Path Finding (MAPF) is a significant problem with pivotal applications in robotics and logistics. The problem involves determining collision-free paths for multiple agents with specific goals in a 2D grid-world environment. Unfortunately, finding optimal solutions for MAPF is an NP-hard problem. Traditional centralized planning approaches are *intractable* for large numbers of agents and *inflexible* when adapting to dynamic changes in the environment. On the other hand, existing decentralized methods utilizing learning-based strategies suffer from two main drawbacks: (1) training takes times ranging from *days to weeks*, and (2) they often tend to exhibit self-centered agent behaviors leading to increased collisions. We introduce a novel approach leveraging the Decision Transformer (DT) architecture that enables agents to learn individual policies efficiently. We capitalize on the transformer's capability for long-horizon planning and the advantages of offline reinforcement learning to *drastically reduce training times to a few hours*. We further show that integrating an LLM (GPT-4o), enhances the performance of DT policies in mitigating undesirable behaviors such as prolonged idling at specific positions and undesired deviations from goal positions. We focus our empirical evaluation on both scenarios with static environments and in dynamically changing environments where agents' goals are altered during inference. Results demonstrate that incorporating an LLM for dynamic scenario adaptation in MAPF significantly enhances the agents' performance and paves the way for more adaptable multi-agent systems.

## 1 INTRODUCTION

Robot swarms are projected to have a significant impact across various sectors, including manufacturing, warehousing, and transportation logistics. Multi-Agent Path Finding (MAPF) encompasses planning methods designed to identify collision-free paths for multiple agents operating within a shared space, while ensuring the optimality of these paths according to specific cost functions (e.g., minimizing path length or travel time). Unfortunately, the MAPF problem is NP-Hard. Much of the literature on MAPF has focused on centralized planning, where a single planner is assumed to have full observability of the world including static and dynamic obstacles. The solution to MAPF computes the plan (which is the set of conflict-free paths for all agents), and is obtained using heuristic search and optimization techniques such as Conflict-Based Search (CBS), M* (a multi-agent version of A*), and ODrM* Sharon et al. (2015); Hart et al. (1968); Wagner & Choset (2011); Ferner et al. (2013). As centralized planning is essentially a constraint solving problem, it is unsurprising that with enough computational resources, optimal collision-free paths for individual agents can be found. However, centralized planning has several disadvantages: (1) it cannot scale to large environments or large number of agents, (2) requires full observability of the agents and the world, (3) produces plans that are not robust to changes in the environment.

In many practical multi-agent setups, the world map is not *a priori* available and each agent is restricted to observing only its own field of view (FOV). In such partially observable environments, autonomous or *decentralized* agents that have the ability to plan their own paths are suitable, and learning-based MAPF methods have been explored. A prominent benchmark is an approach called PRIMAL, where imitation learning (IL) is combined with reinforcement learning (RL), allowing agents to *imitate* centralized planner behaviors while being trained in a decentralized manner Sartoretti et al. (2019). Prioritized Communication Learning method (PICO), an extension of PRIMAL,

incorporates planning priorities and communication topologies into its learning pipeline to improve collision avoidance and boost collaborative behavior Li et al. (2022). On the other hand, Distributed Heuristic Learning with Communication (DHC) tries to achieve this objective by employing graph convolution for agent cooperation, Ma et al. (2021a). Here, each agent operates independently with heuristic guidance provided through potential shortest-path choices. Decision Causal Communication (DCC) enhances DHC by focusing on selective communication. Unlike other methods, DCC enables agents to choose relevant neighbors for communication, minimizing redundancy and overhead Ma et al. (2021b). SCRIMP introduces a differentiable transformer-based communication mechanism, which addresses the challenges posed by partial observations and enhances team-level cooperation Wang et al. (2023). Finally, Confidence-based Auto-Curriculum for Team Update Stability (CACTUS) proposes a reverse curriculum approach that incrementally increases the potential distance between start and goal locations to learn effective policy Phan et al. (2024).

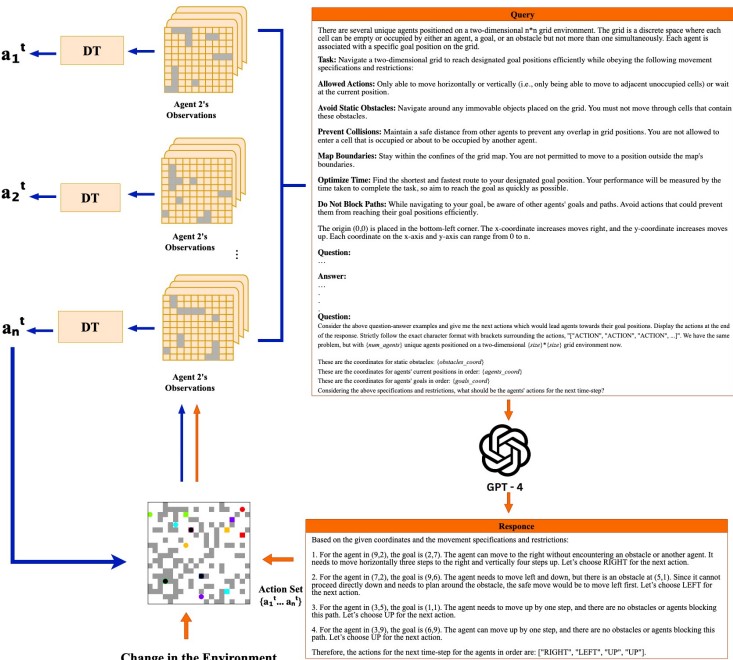

Figure 1: Architecture of our pipeline: blue and orange arrows represent the processes of DT and GPT-4o, respectively.

Most SOTA learning-based MAPF methods integrate complex modules into their pipelines to facilitate communication between agents. This is because the decentralized nature of multi-agent reinforcement learning (MARL) methods often causes agents to act selfishly, disregarding other agents in the environment. However, these intricate pipelines and extensive online interactions with the environment result in *prolonged training times*. In the first part of our paper, we address this challenge by leveraging Decision Transformer-based individual policies using an offline dataset, and show **significant reduction** in training times.

In the second part of our paper, we explore scenarios where LLMs can be used to enhance the performance of DT agents. LLMs have demonstrated the ability to understand and execute instructions for control and embodied tasks in robotics Szot et al. (2023); Yu & Mooney (2022); Liang et al. (2023); Hu et al. (2023); Ahn et al. (2022), coding Liu et al. (2023b); Chen et al. (2021b), strategic planning Wu et al. (2023); Liu et al. (2023a), spatial reasoning Wu et al. (2023), and even planning multi-agent collaboration tasks Li et al. (2023); Zhang et al. (2023); Talebirad & Nadiri (2023). In our experience with DT agents, while they have overall good performance in static environments, there are specific undesirable behaviors that can occur under DT-based policies; here, we use LLMs to guide the agents. Specifically, we observe that LLMs can be useful in situations where an agent gets "boxed in" due to surrounding obstacles, where an agent may oscillate between cells, or where it cannot adapt to changes in the environment. We have integrated GPT-4o into our inference pipeline

Achiam et al. (2023). Additionally, we provide comparative results for GPT-4o and Llama 3.1 on sample environments[1]. Interestingly, we also show that the direct application of LLM-based agents does not yield comparable results to current methods and underperforms the DT-based agents Chen et al. (2024).

Our experimental setup involves two scenarios: one with static environments and one with altered environments. In the first case, DT agents are given a fixed amount of time to finish the episode. GPT-4o is used if at least one of the DT agents fails to complete the task within the allocated time. In the second scenario, we change the goal positions of some agents during the inference and include GPT-4o for only five timesteps after introducing the change in the environment, then we switch the agent policy back to the DT-based policy. In our experiments, we observed that DT agents first explore the previous goal position before moving toward the new goal, whereas GPT-4o directs agents directly toward their new goal. Therefore, alternating DT with GPT-4o for only five timesteps allows for modifications and real-time adjustments, enhancing the adaptability of the agents.

The main contributions of our paper can be summarized as follows:

- We introduce an offline RL approach to train decentralized agents solve the MAPF problem and reduce training time drastically from weeks to several hours while maintaining comparable performance. Table 1 presents a comparison of learning-based MAPF benchmarks with respect to the time consumption of the training process.
- We adopt the decision transformer (DT) architecture to effectively addresses the credit assignment problem in long-horizon MAPF scenarios where episodes can extend to 200-300 time steps with positive rewards given only at the end.
- Our work is among the first to explore the potential of LLMs in MAPF and includes the most comprehensive experiments conducted in randomly generated grid environments.
- We demonstrate that utilizing GPT-4o improves the performance of our DT-based agents in specific navigation tasks within the robotics domain. Our approach highlights the potential of integrating LLMs with RL agents to achieve effective and adaptive behavior in complex environments.

## 2 PRELIMINARIES

### 2.1 PROBLEM SETTING

Our research focuses on a deterministic and partially observable environment where a team of agents operates in a grid world to complete given tasks. Each agent is assigned to move from a start point to an endpoint and can either move to neighboring cells or remain stationary. The goal is achieved by all agents when they complete their tasks while minimizing the total time taken and avoiding collisions with obstacles or other agents.

In real-world applications of this scenario, there may be some unexpected changes in the environment, hence in agents' states. Changes in the position of obstacles and agents' goals are the most common dynamic modifications that require real-time adjustments to the strategy, i.e. adaptation of actions.

### 2.2 MULTI-AGENT PATH FINDING

Our setup constitutes a Decentralized Partially Observable Markov Decision Process (Dec-POMDP) which is a framework used in multi-agent systems where multiple agents must make decisions based on their individual observations of the environment Omidshafiei et al. (2017). The Dec-POMDP is defined by the tuple;

$$(I, S, A, T, \Omega, O, R, \gamma)$$

consisting of $I$: the set of agents, $S$: the set of states, $A = \times_{i \in I} A_i$: the set of joint actions, where $A_i$ is the set of actions available to agent $i$, $T(s' \mid s, a)$: the state transition function that gives the probability of transitioning to state $s'$ from state $s$ after joint action $a$, $\Omega = \times_{i \in I} \Omega_i$: the set of

---

[1]As the primary objective of our paper is not to conduct a comparative analysis of LLMs, we limit our experiments to a selection of representative environments.

Table 1: Effort relative to other benchmarks

| Benchmark | Training Time | Training Episodes |
|---|---|---|
| PRIMAL | $\approx 20$ days | 3.8M |
| DCC | $\approx 1$ day | 128K |
| DT | $\approx 3$ hours | 133K |

joint observations, where $\Omega_i$ is the set of local observations of agent $i$, $O(o \mid s', a)$: the observation function that gives the probability of joint observation $o$ given state $s'$ and joint action $a$, $R(s, a)$: the reward function that gives the immediate reward received after taking joint action $a$ in state $s$, and $\gamma$: the discount factor that determines the importance of future rewards.

In a shared environment, each agent behaves according to its policy $\pi^i(a^i|s) = P(A_t^i = a^i|S_t = s)$, which is the probability distribution over actions given states. At a time step $t$, the joint action $\vec{a}(t) = (a_t^1, \ldots, a_t^N)$, where $N = |I|$, leads to a transition to a new state $s_{t+1}$ according to the transition function $T$, and each agent receives a reward $r_t^i$ according to the reward function $R$. We consider a finite system where each episode plays out for a given $\mathsf{T}$ timesteps.

Additionally, we define $G$ to be the set of goal states $G \subseteq S$ and say that an agent $i$ has reached its goal if $s_t^i \in G$ for some $t \leq \mathsf{T}$. We consider an episode to terminate when all agents have reached their goals or at timestep $\mathsf{T}$, whichever is sooner.

## 2.3 DECISION TRANSFORMER

The Decision Transformer treats offline RL as a sequence modeling problem and learns a return-conditioned policy from an offline dataset Chen et al. (2021a). It has provided a novel perspective to reinforcement learning, and several extensions of this concept have been introduced subsequently Zheng et al. (2022); Lee et al. (2022). In the architecture, embedded tokens of returns, states, and actions are fed into a decoder-only transformer to generate the next tokens autoregressively using a causal self-attention mask. In other words, the model learns the probability of the next token $x_t$ conditioned on previous $K$ tokens, $P_\theta(x_t|x_{t-K<\ldots<t})$, where $K$ is a hyperparameter called *context length*. To achieve this, we consider sequences of the form:

$$\tau^i = (x_1^i, \ldots, x_t^i, \ldots, x_T^i) \quad \text{where} \quad x_t^i = (\hat{R}_t^i, o_t^i, a_t^i)$$

such that $\hat{R}_t^i$ is *return-to-go* (rtg) representing the cumulative rewards from the current time step until the end of the episode, $o_t^i$ is the observation, and $a_t^i$ is the action of agent $i$ at time $t$. Instead of using the discounted rewards as in traditional RL, DT uses rtg so that the model can predict the future actions that achieve the desired return via cross-entropy loss.

We choose DT as the backbone for our method for three major reasons:

- It is an offline RL algorithm that enables training on an offline dataset and reduces training time significantly since it does not require online interaction with the environment during training.
- The transformer architecture effectively addresses the credit assignment problem in long-horizon MAPF scenarios with positive rewards given only at the end.
- DT performs well without extensive reward engineering by conditioning on the desired return.

## 2.4 LARGE LANGUAGE MODELS

Large Language Models (LLMs) are foundation models pre-trained on extensive text corpora to understand, predict, and generate natural language. The introduction of transformers in 2017, particularly due to their key components of positional encodings and self-attention mechanisms, marked a revolutionary advancement in natural language processing Vaswani et al. (2017). Building upon this breakthrough, significant advancements in language models such as BERT, T5, the GPT series,

Llama, and PaLM have extended the capabilities of LLMs and enabled their application to increasingly sophisticated tasks Devlin et al. (2018); Raffel et al. (2020); Radford et al. (2018); Achiam et al. (2023); Touvron et al. (2023); Chowdhery et al. (2023). Notably, these models function as few-shot learners for downstream tasks through prompt engineering without requiring further training Chang et al. (2024).

**In-Context Learning**. (ICL) is an approach utilized by LLMs to handle downstream tasks by conditioning on relevant input-output examples or demonstrations Brown et al. (2020); Dong et al. (2022). This prompting process takes place during the inference stage and does not involve any fine-tuning or updating of the model weights. The provided pairs of instances guide the model to produce accurate responses for the input query. Consequently, a pre-trained LLM model can tackle a wide range of tasks, from translation to question-answering, simply by modifying the examples in the prompt. This flexibility renders ICL a powerful tool for leveraging LLMs in new tasks.

**Chain of Thought**. (CoT) process is an advanced demonstration designing technique for prompt engineering used with LLMs to enhance their problem-solving abilities, particularly for tasks that require complex reasoning including arithmetic, commonsense, and symbolic reasoning tasks Wei et al. (2022). This method involves structuring prompts to guide models through a step-by-step reasoning pathway, ultimately leading to a more informed and accurate output. In CoT prompting, the usual input-output mapping is expanded to include intermediate steps and instead of directly aiming for an answer, the model is prompted with triples: (input, chain of thought, output). Although several advancements in prompt engineering have emerged such as SayCan, ReAct, ToT, and other variations of CoT, we have chosen to utilize CoT in our work due to its simplicity and effectiveness Ahn et al. (2022); Yao et al. (2022; 2024); Wang et al. (2022).

Figure 2: In the query prompt, the MAPF problem, environment, and constraints are first outlined, followed by task-specific in-context examples provided using Chain-of-Thought (CoT). A complete example is provided in the Appendix.

## 3  METHOD

We first create an offline dataset consisting of expert-level trajectories utilized to train the Decision Transformer. The trained DT model is deployed to each agent to generate the next action at each time step. The DT-based agents then navigate towards their goals within the test environment for predetermined time steps of $128, 196, 228$ timesteps for $20, 40, 80$-size environments respectively. Subsequently, GPT-4o is integrated into the pipeline to assist agents that have yet to reach their goals. Environmental information, including the coordinates of static obstacles, the agents' current

positions, and their target destinations, is encoded into a query-prompt. The prompt is then provided to GPT-4o which generates the next set of actions for one time-step. This iterative process continues for five time steps, after which the pipeline reverts to the DT policies to complete the episode. Furthermore, we conducted experiments involving dynamic scenario adaptation by integrating GPT-4o with real-time environmental changes. Detailed descriptions of the methodology and experimental results are presented in the following sections. The overall architecture of our pipeline is depicted in Figure 1.

## 3.1 BUILDING TRAINING DATASET

To generate **expert-level** behavior, we collected trajectories using the ODrM* algorithm, a centralized classical MAPF solver frequently employed in the literature to create expert trajectories for imitation learning. The algorithm was executed on 80 randomly generated grid environments for each combination of varying parameters: $(4, 16, 32, 64)$ agents, grid sizes of $(10, 20, 40, 80)$, and obstacle densities of $(0, 0.1, 0.2)$.

Each agent's path constitutes a distinct trajectory in the training dataset. Given the paper's emphasis on partially observable environments, agents are constrained to observing only their own fields of view (FOV), each of size $10 \times 10$. Observations are represented by four 2-dimensional arrays of shape $(10, 10)$, encoding the following information about their local environments:

- The positions of neighboring agents within the agent's FOV, represented by agent number.
- The position of the agent's own goal.
- The positions of neighbors' goals within the agent's FOV, represented by agent number.
- The positions of obstacles within the agent's FOV, represented by '1's.

Unoccupied cells in the grid are represented by '0's, grid boundaries are treated as obstacles. If the agent's goal lies outside its field of view, the goal is projected onto the edge cell closest to it. If the goal falls within the agent's FOV, it is displayed in the corresponding cell.

At each time step, agents have the option to either wait or move in one of four directions (N/E/S/W) while receiving rewards as outlined: $-0.3$ for moving, $(0/-0.5)$ for waiting (on/off goal), $-5$ for collision, and $+20$ for reaching the goal. We also created a modified version of the dataset in which agents receive an extra reward of $+20$ upon successfully completing an episode, (i.e. all agents reach respective goals). However, the DT model trained on this modified dataset demonstrated inferior performance compared to the model trained on the original version.

Once the trajectories are collected, the dataset undergoes pre-processing to align with the input format required by the DT model. The trajectory for each agent, consisting of return-to-go, observations, and actions, is truncated upon reaching the goal position, as no further rewards are expected beyond that point. With the context length for the DT set to 50, we divided the trajectories into chunks of length 50. For chunks shorter than the specified context length, we applied zero-padding. The final dataset, composed of these chunks, was derived from a total of 133K episodes.

Table 2: Hyperparameters for DT

| Hyperparameter | Value |
|---|---|
| Number of layers | 6 |
| Number of attention heads | 8 |
| Embedding dimension | 128 |
| Batch Size | 128 |
| Context Length K | 50 |
| Return-to-go conditioning | 20 |
| Encoder channels | $8, 16$ |
| Encoder filter sizes | $3 \times 3, 2 \times 2$ |
| Max epochs | 5 |
| Dropout | 0.1 |
| Learning rate | $6 * 10^{-4}$ |
| Adam betas | $(0.9, 0.95)$ |
| Grad norm clip | 1.0 |

## 3.2 TRAINING DECISION TRANSFORMER

For the most part, we retained the original architecture of the Decision Transformer. However, we introduced a few minor modifications, such as replacing the linear layer with a convolutional encoder to process observations of shape $(4, 10, 10)$. Additionally, the hyperparameters listed in Table 2 were employed for training.

### 3.3 PROMPTING GPT-4

We explored various prompting techniques throughout our work; the details and findings presented in the discussion section. The prompt design that yielded the best performance has been integrated into our pipeline; the prompt template is illustrated in Figure 2, and the complete prompt can be found in the Appendix. In the query prompt, we begin by outlining the problem, environment specifications, constraints, and the task. Following this, we provide task-specific in-context examples and pose a similar question generated by our pipeline. To construct these in-context examples, we analyzed environments where the DT model failed to find a path to a goal for at least one agent within T timesteps. Using both a simple and a challenging (failed) scenario, we curated sample question-answer pairs.

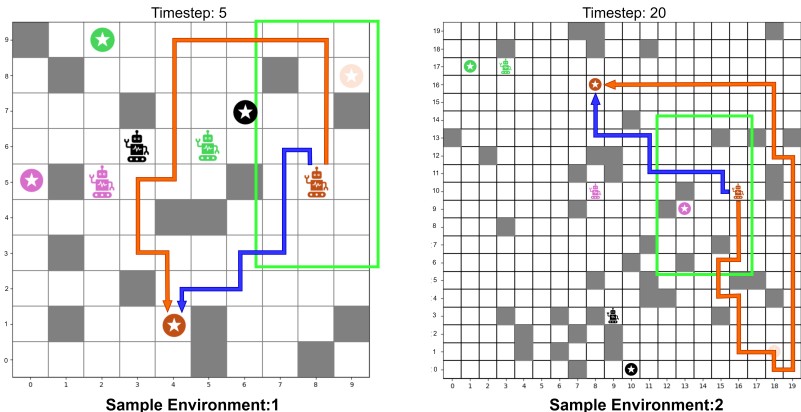

Figure 3: Illustration of GPT-4o's assistance in the event of a goal change. In the first environment, the orange agent initially has its goal at (9,8), which is changed to (4,1) in the $5^{th}$ time step. In the second environment, the orange agent's goal is initially at (18,1), but is altered to (8,16) in the $20^{th}$ time step. The orange arrows depict the path generated by the DT alone, while the blue arrows represent the path taken when decision-making is switched to GPT-4o for five time steps after the goal change, before returning to DT. Green rectangles highlight the five time steps during which GPT-4o and DT make decisions. Without GPT-4o's assistance, DT agents initially explore the region around the previous goal before navigating to the new one. With GPT-4o's assistance, however, the agents can directly navigate toward the new goal.

## 4 EXPERIMENTS

### 4.1 EXPERIMENTAL SETUP

Our experiments are carried out on $n \times n$ grid environments, where $n = \{20, 40, 80\}$ with varying numbers of agents $\{8, 16, 32, 64\}$, and obstacle densities $\{0, 0.1, 0.2\}$.

The results, summarized in the Tables 3 and 4, represent averages across 3 different obstacle density values and 60 environments (combinations of grid size, number of agents), resulting in a total of 720 test environments. We evaluated the performance of the benchmarks and our models using three key metrics: success rate (SR), makespan (MS), and collision rate (CR). The success rate is defined as the ratio of successfully completed episodes (i.e. all agents reach their goals) to the total number of test episodes. The makespan refers to the duration of an episode, specifically the time taken for the last agent to reach its designated goal. Finally, the collision rate is computed as the number of collisions among agents in a successful episode divided by the episode's duration. Collisions occurring in unsuccessful episodes are excluded from this calculation. The model training and experiments have been executed on an NVIDIA Quadro RTX 5000 with 16 GB GPU memory.

Table 3: Comparison of MAPF benchmarks vs our methods: MS, SR, and CR are abbreviations for makespan, success rate, and collision rate, respectively. For makespan and collision rate, lower values (indicated by arrows) means better performance.

| Env Size | # Agents | PRIMAL | | | DCC | | | DT (ours) | | | DT + GPT-4o (ours) | | |
|---|---|---|---|---|---|---|---|---|---|---|---|---|---|
| | | MS $\downarrow$ | SR(%) | CR $\downarrow$ | MS $\downarrow$ | SR (%) | CR$\downarrow$ | MS$\downarrow$ | SR (%) | CR$\downarrow$ | MS$\downarrow$ | SR (%) | CR$\downarrow$ |
| 20x20 | 8 | 50.3 | 94 | 0.34 | **29.6** | **100** | 0.71 | 35.2 | **100** | 0.15 | 32.7 | **100** | **0.13** |
| | 16 | 75.5 | 90 | 1.60 | **42.5** | 96 | 1.52 | 60.9 | 93 | **0.52** | 53.8 | **97** | 0.52 |
| | 32 | 125.5 | 80 | 10.01 | **90.8** | 81 | 4.34 | **90.8** | **87** | **1.55** | 102.6 | 84 | 1.74 |
| 40x40 | 8 | 80.1 | 96 | - | 59.6 | **100** | 0.11 | 57.7 | **100** | 0.04 | **56.9** | **100** | **0.02** |
| | 16 | 115.7 | 89 | - | 71.3 | **100** | 0.60 | 71.6 | 98 | 0.17 | **71.2** | 98 | **0.14** |
| | 32 | 140.3 | 80 | - | **93.5** | **100** | 1.94 | 105.1 | 88 | **0.61** | 104.1 | 88 | 0.65 |
| | 64 | 175.6 | 71 | - | **135.5** | **93** | 11.39 | 160.6 | 76 | 2.29 | 142.7 | 80 | **2.23** |
| 80x80 | 8 | 140.8 | 86 | - | **101.4** | 96 | 0.02 | 115.2 | 97 | 0.02 | 115.8 | **100** | **0.00** |
| | 16 | 180.7 | 81 | - | 122.2 | **96** | 0.38 | 123.9 | 92 | 0.05 | **118.1** | 90 | **0.04** |
| | 32 | 230.3 | 75 | - | **132.9** | **96** | 0.59 | 146.6 | 88 | 0.27 | 144.5 | 90 | **0.23** |
| | 64 | 250.4 | 57 | - | **159.6** | **91** | 2.75 | 183.1 | 64 | 0.76 | 177.8 | 72 | **0.69** |

## 4.2 RESULTS

**Stationary Environments**. The results presented in Table 3 indicate that DT-based agents demonstrate strong performance (i.e. lower CR, lower MS, and higher SR) compared to SOTA MAPF models[2]. We draw the following conclusions from our experiments:

1. DT-based agents consistently outperform PRIMAL across all evaluation metrics.
2. Our method surpasses DCC in terms of collision rates, indicating that our agents exhibit safer behavior. This is particularly significant since collisions among agents in real-world scenarios, such as warehouses, can lead to substantial costs and safety hazards.
3. Our methods outperform DCC in terms of success rate in 20x20 environments, indicating superiority in smaller settings.
4. Further, the integration of GPT-4o with DT agents enables navigation along even shorter and safer paths compared to DT agents alone. Although the primary intent of incorporating a LLM was to handle real-time environmental changes, the results reveal that it also offers considerable benefits in static environments.

**Dynamic Scenario Adaptation**. For $20, 40, 80$ size environments, we modify the environment once at 15th, 30th, and 50th timesteps respectively. We conduct our experiments according to two difficulty levels; altering the goals of .25 of the agents and .5 of the agents in the environments during inference. Based on the results presented in Table 4, we can draw the following conclusions:

1. The integration of GPT-4o reduces makespan across most environmental settings. This reduction is particularly significant when the new goal location is in the opposite direction to the agents' prior trajectory. As illustrated in Figure 3, DT requires several timesteps to comprehend the goal change, often exploring areas near the previous goal location before adjusting its direction. In contrast, when the LLM-based suggestions are introduced concurrently with the dynamic goal change, agents immediately reorient towards their new goal location – this significantly reduces the makespan.
2. The success rates achieved by DT and LLM collaboration are equal to or surpass those of the DT alone in most environmental settings. Notably, when we alter the goal positions of half of the agents, the advantages of LLM guidance become particularly evident in complex environments characterized by larger sizes and a greater number of agents.
3. The reduction in CR in static environments by using an LLM also extends to dynamic environments; DT+GPT-4o consistently achieves lower collision rates, leading to safer agent behavior.

These findings demonstrate that LLM-assisted DT-based agents are highly effective for real-time adaptations and offer significant advantages in safety-critical environments.

---

[2]We do not report CR for PRIMAL in 40x40, 80x80 grid sizes as they are not available in the literature; training time for PRIMAL is very high ($\approx$ 3 weeks), so repeating experiments to obtain the CR is infeasible.

Table 4: Impact of GPT-4o on dynamic environments: MS, SR, and CR are abbreviations for makespan, success rate, and collision rate, respectively. For makespan and collision rate, lower values (indicated by arrows) means better performance.

| Env Size | # Agents | DT (1/4) | | | DT+GPT-4o (1/4) | | | DT (1/2) | | | DT+GPT-4o (1/2) | | |
|---|---|---|---|---|---|---|---|---|---|---|---|---|---|
| | | MS↓ | SR(%) | CR↓ | MS↓ | SR (%) | CR↓ | MS↓ | SR (%) | CR↓ | MS↓ | SR (%) | CR↓ |
| 20x20 | 8 | 48.5 | 95 | 0.21 | 44.7 | 97 | 0.11 | 48.9 | 96 | 0.14 | 52.8 | 97 | 0.08 |
| | 16 | 68.5 | 85 | 0.48 | 68.1 | 88 | 0.51 | 71.5 | 85 | 0.49 | 74.2 | 95 | 0.37 |
| | 32 | 100.1 | 73 | 1.68 | 101.1 | 71 | 1.64 | 112.1 | 78 | 1.68 | 102.2 | 75 | 1.55 |
| 40x40 | 8 | 80.9 | 98 | 0.02 | 77.0 | 100 | 0.02 | 92.4 | 96 | 0.04 | 91.8 | 100 | 0.03 |
| | 16 | 97.2 | 98 | 0.15 | 92.4 | 98 | 0.14 | 104.4 | 92 | 0.13 | 97.1 | 98 | 0.12 |
| | 32 | 126.4 | 82 | 0.67 | 125.5 | 82 | 0.50 | 120.2 | 75 | 0.48 | 120.2 | 82 | 0.54 |
| | 64 | 162.9 | 60 | 2.03 | 150.7 | 61 | 1.82 | 172.7 | 60 | 2.18 | 161.5 | 64 | 1.82 |
| 80x80 | 8 | 133.0 | 94 | 0.01 | 128.5 | 94 | 0.00 | 147.9 | 93 | 0.02 | 145.4 | 92 | 0.01 |
| | 16 | 144.4 | 92 | 0.03 | 141.0 | 97 | 0.03 | 146.0 | 90 | 0.04 | 150.1 | 94 | 0.07 |
| | 32 | 155.5 | 82 | 0.18 | 158.0 | 81 | 0.17 | 171.7 | 78 | 0.16 | 166.3 | 85 | 0.24 |
| | 64 | 182.4 | 66 | 0.59 | 176.3 | 62 | 0.72 | 169.8 | 63 | 0.70 | 183.0 | 62 | 0.53 |

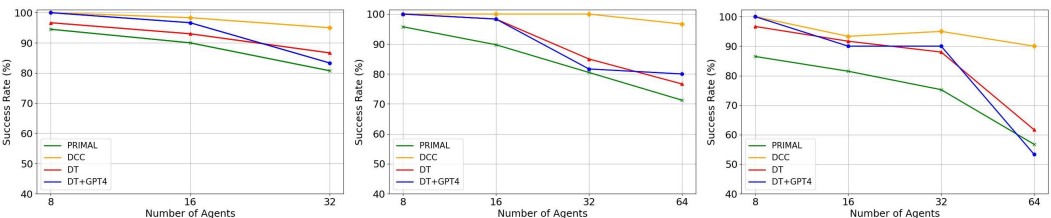

Figure 4: Success rates for 20, 40, 80-size environments in static environments.

## 4.3 COMPARATIVE ANALYSIS

**Centralized vs. Decentralized Approaches in MAPF**. Centralized MAPF algorithms such as ODrM* cannot scale to larger numbers of agents due to inherent intractability of the MAPF problem, and as the number of agents increases, the central planner must consider a growing number of potential interactions and paths. Assumptions of full map observability can be unrealistic in environments with sensory limitations or privacy constraints. Decentralized learning-based agents can make optimal decisions based on partial and local observations, which allows the model to be replicated across many agents. Constraint-solving based centralized planning methods must also perform expensive re-planning for all agents whenever any environmental change occurs. These issues of scalability, full observability, and adaptability associated with centralized methods can make them impractical in many MAPF settings. In our experiments, we thus skip comparing to centralized methods as it is not an appropriate comparison.

**Offline vs. Online RL Approaches in MAPF**. We show that incorporating offline RL in MAPF (through the DT architecture) is an effective learning strategy that yields performance comparable to other learning-based methods that require online interaction with the environment during training. Table 1 highlights the significant reduction in required effort and our success in eliminating the necessity for real-time interaction with environments during training. Notably, our model does not experience the distributional shift issues that challenge offline RL algorithms when tested in new environments. By utilizing a dataset consisting of a broad range of samples from randomly generated grid environments, we mitigate the risk of distributional shift (analogous to Tobin et al. (2017)).

## 4.4 DISCUSSION

**Decision Transformer in Multi-Agent Setting**. The effectiveness of treating offline reinforcement learning as a sequence modeling problem and leveraging the transformer architecture has been demonstrated by the Decision Transformer. Our findings indicate that the Decision Transformer

Table 5: Comparison of LLMs on sample environments

| | | GPT-4o | | Llama-3.1 | |
|---|---|---|---|---|---|
| *Env Size* | *# Agents* | *Makespan* ↓ | *Success Rate* (%) | *Makespan* ↓ | *Success Rate* (%) |
| 10x10 | 8 | 46.7 | **80** | **36.1** | 60 |
| 20x20 | 8 | **51.4** | **50** | 52.5 | 40 |

performs well in a multi-agent RL setting when agents are trained using a decentralized approach. Notably, this application achieves good performance without necessitating any modifications to the original model.

**LLMs' Efficiency in Real-Time Adjustments**. The environments in which agents operate can be dynamic with obstacles being added or removed and goal locations altered in real-time. We observe that the DT agents performs well in response to changes in obstacles, which makes LLM assistance unnecessary in such cases. Agents quickly adapt and navigate around newly introduced obstacles. However, DT agents struggle when goal locations change as indicated in Figure 3. Initially, they move toward their previous goal positions and explore those areas before eventually redirecting to the new goals. This delay is critical, as it increases both time and energy consumption in practical applications. Our trials using GPT-4o for $3, 5, 7,$ and $10$ timesteps indicate that guidance from GPT-4o for 5 timesteps yields the best performance. By incorporating an LLM, agents can take more efficient actions, which reduces unnecessary movements and prevents repetitive behaviors.

**Prompt Engineering**. Prompt engineering is crucial for harnessing the potential of LLMs in complex reasoning tasks. We conducted several trials targeting two main objectives: optimizing the prompt itself and selecting appropriate in-context examples. To identify the most effective prompt, we utilized GPT-4o iteratively. We first described the problem, environment, and task, then asked GPT-4o to rephrase the problem and setup in its own words. This process was followed by tests in sample environments with corrective feedback provided to GPT-4o. This cycle continued until no further improvements in performance were observed.

Additionally, to determine which in-context examples produced the best outcomes, we tested various sets of examples: one set of simple example pairs, one set of difficult example pairs, a set arranged by increasing difficulty (analogous to curriculum learning), and sets with varying numbers of examples. For challenging environments, we analyzed agent failures and provided step-by-step reasoning for correct actions, similar to Chain-of-Thought (CoT) prompting. We observed that a set of four example pairs ordered by increasing difficulty was the most effective in our problem setting. Figures 5 and 6 in the Appendix illustrate the environments used in our final testing experiments.

## 5 CONCLUSION

Despite their success in diverse areas, LLMs may hallucinate, i.e., yield outputs that deviate from factual accuracy or contextual relevance, particularly in long-horizon reasoning and planning problems Kambhampati (2024). This research attempts to harness the capabilities of LLMs within MAPF and points out contexts wherein the utilization of the models addresses specific challenges.

**Limitations & Future Work**. In this paper, we opted for textual inputs because LLMs are still largely unexplored within the MAPF literature. Replicating this approach using visual inputs and Visual Language Models (VLMs) presents a promising direction for future research. Furthermore, our trials with OpenAI's recent o1-preview (Strawberry) model demonstrated success in environments where GPT-4o failed. This highlights the rapid advancements in LLM capabilities. Given these developments, we believe that the integration of LLMs into MAPF methods is promising.

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

## A APPENDIX

### A.1 IN-CONTEXT EXAMPLES

Figure 5: $10 \times 10$ Simple sample environment used to create in-context examples

After running multiple experiments, we have observed that using one simple and one difficult environments to create in-context examples gives the best performance of LLMs. Figures 5 and 6 are the environments used in our final testing experiments.

### A.2 FULL PROMPT

There are several unique agents positioned on a two-dimensional n*n grid environment. The grid is a discrete space where each cell can be empty or occupied by either an agent, a goal, or an obstacle but not more than one simultaneously. Each agent is associated with a specific goal position on the grid.

**Task:**. Navigate a two-dimensional grid to reach designated goal positions efficiently while obeying the following movement specifications and restrictions:

| (0,9) | (1,9) | (2,9) | (3,9) | (4,9) | (5,9) | (6,9) | (7,9) | (8,9) | (9,9) |
|---|---|---|---|---|---|---|---|---|---|
| (0,8) | (1,8) | (2,8) | (3,8) | (4,8) | (5,8) | (6,8) | (7,8) | (8,8) | (9,8) |
| (0,7) | (1,7) | (2,7) | (3,7) | (4,7) | (5,7) | (6,7) | (7,7) | (8,7) | (9,7) |
| (0,6) | (1,6) | (2,6) | (3,6) | (4,6) | (5,6) | (6,6) | (7,6) | (8,6) | (9,6) |
| (0,5) | (1,5) | (2,5) | (3,5) | (4,5) | (5,5) | (6,5) | (7,5) | (8,5) | (9,5) |
| (0,4) | (1,4) | (2,4) | (3,4) | (4,4) | (5,4) | (6,4) | (7,4) | (8,4) | (9,4) |
| (0,3) | (1,3) | (2,3) | (3,3) | (4,3) | (5,3) | (6,3) | (7,3) | (8,3) | (9,3) |
| (0,2) | (1,2) | (2,2) | (3,2) | (4,2) | (5,2) | (6,2) | (7,2) | (8,2) | (9,2) |
| (0,1) | (1,1) | (2,1) | (3,1) | (4,1) | (5,1) | (6,1) | (7,1) | (8,1) | (9,1) |
| (0,0) | (1,0) | (2,0) | (3,0) | (4,0) | (5,0) | (6,0) | (7,0) | (8,0) | (9,0) |

Figure 6: $10 \times 10$ Difficult sample environment used to create in-context examples

**Allowed Actions:.** Only able to move horizontally or vertically (i.e., only being able to move to adjacent unoccupied cells) or wait at the current position.

**Avoid Static Obstacles:.** Navigate around any immovable objects placed on the grid. You must not move through cells that contain these obstacles.

**Prevent Collisions:.** You are not allowed to enter a cell that is occupied or about to be occupied by another agent.

**Map Boundaries:.** Stay within the confines of the grid map. You are not permitted to move to a position outside the map's boundaries.

**Optimize Time:.** Find the shortest and fastest route to your designated goal position. Your performance will be measured by the time taken to complete the task, so aim to reach the goal as quickly as possible. Prioritize taking actions that move agents directly to the goal. If both directions are blocked by obstacles or agents, try to move around if they are both obstacles or wait for a time step and let the agent move away then start the movement in the next timestep.

**Do Not Block Paths:.** While navigating to your goal, be aware of other agents' goals and paths. Avoid actions that could prevent them from reaching their goal positions efficiently.

The origin (0,0) is placed in the bottom-left corner. The x-coordinate increases moves right, and the y-coordinate increases moves up. Each coordinate on the x-axis and y-axis can range from 0 to n.

**Question:.** These are the coordinates for static obstacles: [(4,5),(4,4),(4,3)], meaning no agent will be permitted to move into any of these cells. These are the coordinates for agents' current positions in order: [(0,7), (3,1), (6,3), (5,7)] These are the coordinates for agents' goals in order: [(1,9), (5,3), (3,4), (7,7)]

Considering the above specifications and restrictions, what should be the agents' actions for the next time-step? Give me actions in order for all 4 agents.

**Answer:.** For the agent in (0,7), the distance between the agent and its goal can be decomposed into 1 step to the right and 2 steps to the up. There are no agents or static obstacles adjacent to the agent. If we move up, the new cell is (0,8) which is not occupied by another agent or static obstacles and brings the agent closer to its goal at (1,9). Hence, let's choose UP for the next action.

For the agent in (3,1), the distance between the agent and its goal can be decomposed into 2 steps to the right and 2 steps to the up. There are no agents or static obstacles adjacent to the agent. If we move right, the new cell (4,1) is not occupied by another agent or static obstacles also brings the agent closer to its goal position at (6,3). Hence, let's choose RIGHT for the next action.

For the agent in (6,3), the distance between the agent and its goal can be decomposed into 3 steps to the left and 1 step to the up. There are no agents or static obstacles adjacent to the agent. But there are static obstacles on the shortest path to its goal, so the agent should consider long-term planning. Considering this, the distance between the agent and its goal can be decomposed into 1 step to the

down, 3 steps to the left and 2 steps to the up. If we move down, the agent moves to (6,2) which brings the agent closer to the goal position at (3,4). Let's choose DOWN for the next action.

For the agent in (5,7), the distance between the agent and its goal can be decomposed into 2 steps to the right. There are no agents or static obstacles adjacent to the agent. If we move right, the new cell is (6,7) which is not occupied by another agent or static obstacles and brings the agent closer to its goal at (7,7). Hence, let's choose RIGHT for the next action.

Hence, the answer is ["UP", "RIGHT", "DOWN", "RIGHT"].

**Question:** These are the coordinates for static obstacles: [(4,5),(4,4),(4,3)], meaning no agent will be permitted to move into any of these cells. These are the coordinates for agents' current positions in order: [(0,8), (4,1), (6,2), (6,7)] These are the coordinates for agents' goals in order: [(1,9), (5,3), (3,4), (7,7)]

Considering the above specifications and restrictions, what should be the agents' actions for the next time-step? Give me actions in order for all 4 agents.

**Answer:** For the agent in (0,8), the distance between the agent and its goal can be decomposed into 1 step to the right and 1 step to the up. There are no agents or static obstacles adjacent to the agent. If we move up, the new cell is (0,9) which is not occupied by another agent or static obstacles and brings the agent closer to its goal at (1,9). Hence, let's choose UP for the next action.

For the agent in (4,1), the distance between the agent and its goal can be decomposed into 1 step to the right and 2 steps to the up. There are no agents or static obstacles adjacent to the agent. If we move right, the new cell is (5,1) which is not occupied by another agent or static obstacles and brings the agent closer to its goal at (5,3). Hence, let's choose RIGHT for the next action.

For the agent in (6,2), the distance between the agent and its goal can be decomposed into 3 steps to the left and 2 steps to the up. There are no agents or static obstacles adjacent to the agent. If the agent goes up now, there may be static obstacles on the shortest path to its goal, the agent should consider long-term planning. Considering this, the agent should complete the left steps first, then go up. If the agent choses left, the new position will be (5,2) which brings the agent closer to its goal at (3,4). Let's choose LEFT for the next action.

For the agent in (6,7), the distance between the agent and its goal can be decomposed into 1 step to the right. There are no agents or static obstacles adjacent to the agent. If we move right, the new cell is (7,7) which is not occupied by another agent or static obstacle and brings the agent to its goal at (7,7). Hence, let's choose RIGHT for the next position.

Hence, the answer is ["UP", "RIGHT", "LEFT", "RIGHT"]

**Question:** These are the coordinates for static obstacles: [(0,8), (1,8), (2,8), (0,4), (1,3), (2,5), (2,4), (2,3), (2,2), (3,6), (3,2), (3,1), (4,1), (8,6), (8,5), (8,4), (8,3), (8,2), (8,1), (8,0), (9,6)], meaning no agent will be permitted to move into any of these cells. These are the coordinates for agents' current positions in order: [(0,9), (2,9), (0,2), (5,2), (7,2)] These are the coordinates for agents' goals in order: [(3,9), (1,9), (0,5), (9,2), (8,2)]

Considering the above specifications and restrictions, what should be the agents' actions for the next time-step? Give me actions in order for all 5 agents.

**Answer:** For the agent in (0,9), the distance between the agent and its goal can be decomposed into 3 steps to the right. There is an obstacle adjacent to this agent at (0,8) so we will not be able to move down. The agent can move to the right. If we move right, the new cell is (1,9) which is not currently occupied by any agents or static obstacles and brings the agent closer to its goal. Let's choose RIGHT for the next action.

For the agent in (2,9), the distance between the agent and its goal can be decomposed into 1 step to the left. There are obstacles adjacent to the agent at (2,8) and an agent at (1,9), thus preventing this agent from moving to the down and to the left. The agent that is blocking needs to move right to arrive at its goal. To not block that agent's path, we move to right now, the new cell is (3,9) which is not currently occupied by any agents or static obstacles and then continue to the goal once the agent has moved away. If the agent moves right, it will be in the (3,9) cell which is currently unoccupied. Let's choose RIGHT for the next action.

For the agent in (0,2), the distance between the agent and its goal can be decomposed into 3 steps to the up. There are no agents or static obstacles adjacent to the agent, however, if the agent goes up, there is a static obstacle at (0,4) on its path to the goal at (0,5). Considering the static obstacles on the agent's path to its goal, the agent needs to move towards an obstacle-free path to its goal, so it needs to go around the obstacles. Going down accomplishes this and (0,1) is currently empty. Let's choose DOWN for the next action.

For the agent in (5,2), the distance between the agent and its goal can be decomposed into 4 steps to the right. There are no agents or static obstacles adjacent to the agent. If we move right, the new cell is (6,2) which is not currently occupied by any agents or static obstacles and moves the agent closer to its goal at (9,2). Let's choose RIGHT for the next action.

For the agent in (7,2), the distance between the agent and its goal can be decomposed into 1 step to the right. However, if we move right, the new cell will be (8,2) which is blocking another goal in (9,2) which is the goal for the agent currently in 8(5,2). To not block another agent's path, we move up or down now. If we move up, the new cell is (7,3) which is not currently occupied by any agents or static obstacles. Let's choose UP for the next action.

Hence, the answer is ["RIGHT", "RIGHT", "DOWN", "RIGHT", "UP"].

**Question:** These are the coordinates for static obstacles: [(0,8), (1,8), (2,8), (0,4), (1,3), (2,5), (2,4), (2,3), (2,2), (3,6), (3,2), (3,1), (4,1), (8,6), (8,5), (8,4), (8,3), (8,2), (8,1), (8,0), (9,6)], meaning no agent will be permitted to move into any of these cells. These are the coordinates for agents' current positions in order: [(1,9), (3,9), (0,1), (6,2), (7,3)] These are the coordinates for agents' goals in order: [(3,9), (1,9), (0,5), (9,2), (8,2)]

Considering the above specifications and restrictions, what should be the agents' actions for the next time-step? Give me actions in order for all 5 agents.

**Answer:** For the agent in (1,9), the distance between the agent and its goal can be decomposed into 2 steps to the right. There is a static obstacle at (1,8) meaning the agent is not allowed to move down. The agent can move to the right. If we move right, the new cell is (2,9) which is not currently occupied by any agents or static obstacles and moves the agent closer to its goal at (3,9). Let's choose RIGHT for the next action.

For the agent in (3,9), the distance between the agent and its goal can be decomposed into 2 steps to the left. However, if we move left, the new cell is (2,9) which will be occupied by another agent. To not block that agent's path, we move down now, the new cell is (3,8) which is not currently occupied by any agents or static obstacles. Let's choose DOWN for the next action.

For the agent in (0,1), the distance between the agent and its goal can be decomposed into 4 steps to the up. There are no agents or static obstacles adjacent to the agent. However, if the agent goes up, there is a static obstacle on its path to the goal at (0,4). Considering the static obstacles on the agent's path to its goal. The agent needs to move towards an obstacle-free path to its goal, so it needs to go around the obstacles. Let's choose UP for the next action which moves the agent to (0,2) which is closer to the goal but be sure to remember that we have to move around the obstacle at (0,4).

For the agent in (6,2), the distance between the agent and its goal can be decomposed into 3 steps to the right. There are no agents or static obstacles adjacent to the agent. If we move right, the new cell is (7,2) which is not currently occupied by any agents or static obstacles and moves the agent closer to the goal at (9,2). Let's choose RIGHT for the next action.

For the agent in (7,3), the distance between the agent and its goal can be decomposed into 1 step to the right and 1 step to the down. There is a static obstacles adjacent to the agent at (8,3) meaning the agent is not allowed to move to the right. There is also an agent at (7,2) meaning the agent is not allowed to move down. Hence, let's choose WAIT for the next action and once the agent below moves, we can move down and get closer to the goal at (8,2).

Hence, the answer is ["RIGHT", "DOWN", "DOWN", "RIGHT", "WAIT"].

**The query question below is generated by our pipeline:**

**Question**:

Consider the above question-answer examples and give me the next actions which would lead agents towards their goal positions. Display the actions at the end of the response. Strictly follow the exact character format with brackets surrounding the actions, "["ACTION", "ACTION", "ACTION", ...]".

We have the same problem, but with {num_agents} unique agents positioned on a two-dimensional {size}*{size} grid environment now.

These are the coordinates for static obstacles: {obstacles_coord}
These are the coordinates for agents' current positions in order: {agents_coord}
These are the coordinates for agents' goals in order: {goals_coord}

Considering the above specifications and restrictions, what should be the agents' actions for the next time-step?

Take into consideration that we have to move towards the goal, thus we shouldn't be waiting if we can take an action that gets an agent closer towards its goal. If an action would collide an agent into a static obstacle, prefer actions that aim to move around that obstacle while also making sure those directions wouldn't collide into another obstacle. Give me the set of actions in order for all {num_agents} agents.

