# OpenReview forum: "Multi-Agent Path Finding via Decision Transformer and LLM Collaboration"
_ICLR.cc/2025/Conference — Submitted to ICLR 2025_

### Official Review · Reviewer_bzvf · 2024-10-29

**Soundness:** 2
**Presentation:** 3
**Contribution:** 2
**Rating:** 3
**Confidence:** 5

**Summary:**

The paper addresses the Multi-Agent Path Finding problem using a Decision Transformer architecture combined with a Large Language Model to improve decentralized learning efficiency and agent coordination. The proposed approach aims to reduce training times significantly and improve adaptability in both static and dynamic environments.

**Strengths:**

The paper includes well-designed visual aids, which effectively illustrate the experimental setups and outcomes.

**Weaknesses:**

The primary contribution is the combination of DT with LLMs, both well-established methods, but the motivation behind integrating these two specific algorithms is not clearly justified. The combination feels somewhat arbitrary, as each component could theoretically be combined with other methods without unique synergy.

Incorporating LLMs potentially introduces global information into the system, which conflicts with the decentralized nature of the approach. This inclusion may undermine the paper's objective of maintaining a decentralized framework.

While the study asserts that offline reinforcement learning can reduce training times, the experimental setup is too simplistic, relying only on simulated environments. Evaluations in real-world settings would provide a stronger validation of the method's claims and its practical applicability in realistic, complex scenarios.

The paper lacks clarity on the mechanism for assigning credit to each agent. The approach claims to be decentralized, however the training relies on a joint reward (e.g., collision penalty). It remains unclear how this joint reward is allocated to each agent individually.

**Questions:**

I have no further questions.

---

> ### Author Response · Authors · 2024-11-22
> **Some Remarks**
>
> Our primary motivation was to leverage these two well-established methods and their complementary strengths to address challenges in MAPF. As you mentioned, each component could indeed be combined with other methods, which adds flexibility to our work and opens up future research directions for the MAPF community.
>
> Our pipeline is built upon DT, which maintains decentralization and operates under partial observability. As a reminder, GPT-4o is utilized specifically for dynamic environments, not static ones. Also, GPT-4o provides suggestions that are used for only 5 time steps. While you could say that the decentralized assumption is violated during these intervening 5 time steps, the agents resume decentralized navigation with DT afterward. If we use only DT without the LLM, we achieve a purely decentralized solution (which takes longer to reach the goal). We reported the results of DT combined with GPT-4o for static environments as some reviewers might ask for this information.
>
> We are puzzled by the comment about "relying only on simulated environments" as the entire RL literature is based on simulated environments. Could you please elaborate on what you meant by this? Our experimental setup and test environments are consistent with those used by benchmark methods in the MAPF literature. This ensures a fair and meaningful comparison.
>
> The collision penalty does not contradict the decentralized approach of our paper. Each agent’s trajectory contributes separately to the dataset, as we explained in the paper. When a collision occurs between two agents, both agents receive the collision reward/penalty independently. This is the standard practice in decentralized learning-based MAPF literature. Please refer to following work:
>
> https://arxiv.org/pdf/2106.11365
>
> https://arxiv.org/abs/1809.03531
>
> https://arxiv.org/pdf/2109.05413
>
> https://arxiv.org/pdf/2303.00605

---

### Official Review · Reviewer_jkSV · 2024-11-03

**Soundness:** 2
**Presentation:** 3
**Contribution:** 2
**Rating:** 5
**Confidence:** 5

**Summary:**

The paper is devoted to the use of offline-RL methods and language models to solve the multi-agent pathfinding problem. The authors use the well-known Decision Transformer architecture, which is trained on a pre-collected dataset from the ODrM* heuristic planner. At some point in time, the agents' policy switches from a DT-based policy, to actions within 5 steps, which is generated by GPT-4 in a centralized manner. The authors proposed a special chain-of-thoughts style prompt to resolve conflicts that arise most often when agents switch goals. The authors conducted experiments with a stationary setting and compared them with PRIMAL and DCC methods, showing some advantages mainly in reducing collisions. For changing goals, a quality improvement is shown by using GPT-4.

**Strengths:**

The authors consider an important formulation of the MAPF problem with changing goals and are among the first to use a combination of DT and LLM to generate agent actions. The paper as a whole is written in easy-to-understand language, and the idea of the method and the experiments are very clear.

**Weaknesses:**

The main disadvantage of the paper is the author's lack of familiarity with the latest benchmarks in the field of learning MAPF and weak positioning of the paper from the point of view of the MAPF community. At the same time, it should be noted that from the point of view of using neural network models and representation models, the novelty of the work is not traceable, so it can only be evaluated in terms of its contribution to the general pool of modern works on learning MAPF. Several specific remarks:
1. Since the prompt for LLM describes entirely the map and information about other agents (as can be understood from the text), the proposed approach is no longer fully decentralized and partially observable. This is analogous to calling a centralized scheduler like ODrM* at some point in time and taking action from it five steps ahead. This aspect, which violates the original Dec-POMDP problem statement, should at least be discussed in detail, and comparing DT+LLM with DT+ODrM* or DT+DCC would improve the work.
2 The authors, by introducing agent communication via LLM, should ideally consider and compare with other work using explicit agent communication with language models, including for the MAPF task.
3. The authors point out that centralized planners are not scalable to a large number of agents. However, first, the authors themselves consider only 64 agents, which is also small, and second, there are modern centralized planners such as LaCaM [1] that scale up to 1K agents.
4. The authors use a 5-year-old PRIMAL method as their main baseline. At a minimum, the authors need to include in the comparison the SCRIMP and CACTUS they mentioned, which are seriously better than PRIMAL in classical MAPF tasks.
5. The authors use a rather simple formulation of the problem with nonstationary conditions, although there are several papers [2], in which the problem of nonstationarity in MAPF has already been considered. Also, the formulation used by the authors is similar to the life-long MAPF problem, which has its own set of methods, such as PRIMAL2 [5].
6. The authors use custom maps and low obstacle density. Although there is a large set of maps and benchmarks (MovingAI [3], POGEMA[4]) that make sense to use as the main dataset for experiments. So that comparisons can be more correct and complete.
7. Emphasizing a significant reduction in training time cannot be considered correct. First, the number of parameters in each of the models (PRIMAL, DCC, DT) should be taken into account. It is possible that somewhere the work with large maps (like 128 by 128) was assumed and neural network models are more redundant than those used by the authors. Secondly, training on demonstrations is a significant simplification of the optimization problem and it is quite natural that the convergence of the process here will be orders of magnitude faster. The use of expert data, which also needs to be collected, is a significant competitive advantage.

[1] https://ojs.aaai.org/index.php/AAAI/article/download/26377/26149
[2] https://peerj.com/articles/cs-1056
[3] https://movingai.com/
[4] https://pypi.org/project/pogema/
[5] https://arxiv.org/pdf/2010.08184

**Questions:**

1. How did the authors account for the variable number of agents to tokenize actions and observations when training DT?
2. Would the results be significantly worse if only the observations seen by each agent were passed to the LLM and decentralization was not violated?

---

> ### Author Response · Authors · 2024-11-22
> **Thank you for your detailed review of our paper**
>
> We appreciate your insights and the time you've taken to provide constructive feedback. We address your concerns and questions below.
>
> Weaknesses:
>
> We disagree with the assessment that we are unfamiliar with the latest learning-based Multi-Agent Path Finding (MAPF) methods. We have cited several recent works, including DHC (2021), DCC (2021), PICO (2022), SCRIMP (2023), and CACTUS (2024).
>
> Previous learning-based methods for MAPF have utilized transformer architectures within imitation learning or online reinforcement learning with hand-crafted reward functions. By leveraging DT, we achieved both introducing offline RL to MAPF literature and eliminating the need for extensive reward engineering. We are the first to apply DT in a multi-agent setting. Additionally, we have explored scenarios where LLMs can be utilized. To our knowledge, our contributions involving DT and LLMs are among the first in the field and represent novel advancements.
>
> 1-)  Our pipeline is built upon DT, which maintains decentralization and operates under partial observability. As a reminder, GPT-4o is utilized specifically for dynamic environments, not static ones. Also, GPT-4o provides suggestions that are used for only 5 time steps. While you could say that the decentralized assumption is violated during these intervening 5 time steps, the agents resume decentralized navigation with DT afterward. If we use only DT without the LLM, we achieve a purely decentralized solution (which takes longer to reach the goal). We reported the results of DT combined with GPT-4o for static environments as some reviewers might ask for this information.
>
> 2-)  Our experiments involved up to 64 agents, consistent with prior learning-based MAPF studies we compared against, such as PRIMAL and DCC. Thanks for the information. We will check out LaCaM and its experiment setup.
>
> 3-) We compare our results against PRIMAL because most papers using a learning-based approach also compare against PRIMAL, making it a common benchmark. By comparing against PRIMAL, we provide context for our method relative to other approaches that report their performance against this benchmark. We chose one widely-used benchmark and one the best-performing method (to our knowledge). We specifically selected these two methods to provide a comprehensive insight into our method. As proof of this, CACTUS reported relative performance against PRIMAL, while SCRIMP used DHC (a previous work of the authors of DCC) for comparison.
>
> For a quick comparison;
>
> - CACTUS’s success rates drops to ~%75 and ~%60 for 64 agents in 40-sized and 80-sized environments respectively. Our success rates are %76 and %64 for DT alone, %80 and %72 for DT+GPT-4o in these settings.
>
> - SCRIMP did not report any results for 80-size environments. It only reports results for (10, 30, 40) sized environments. For 40-sized environments, its success rate is  ~%90. This suggests that SCRIMP is likely optimized for smaller-sized environments.
>
> We can share the detailed comparison results in the revised paper if needed.
>
> 4-) From the “learning” perspective, stationary vs. non-stationary environments or changing goal positions during inference vs. lifelong MAPF represent very different problems. We chose this setting because it aligns with many other learning-based MAPF studies. Specifically, DHC, DCC, PICO, SCRIMP, CACTUS, as well as LaCAM and PRIMAL2 that you cited, all focus on stationary environments. This is the first paper to apply DT to MAPF. For future studies, we will consider your suggestions.
>
> 5-) We used similar experimental setups and environments as the benchmarks we compared against. However, we can conduct additional experiments and include the results in the revised paper.
>
> 6-) Training time is significantly reduced because, while other methods use complex modular architectures to address various challenges, we employ a compact DT architecture. Many MAPF methods, such as PRIMAL, utilize 'training on demonstrations' through imitation learning, but they do not converge quickly.
>
> -PRIMAL takes almost 3 weeks to train with 13M parameters.
>
> -DCC takes a day to train with 1M parameters.
>
> -DT takes 3 hours to train with 1.3M parameters.
>
>
> Questions:
>
> 1-) Each agent's trajectory contributes separately to the dataset, with each trajectory divided into tokens. A token is defined as a tuple of (R_t, o_t, a_t), as described in Section 2.3. This design ensures scalability and allows the DT to handle varying numbers of agents.
>
> 2-) This is an insightful suggestion. (It would require a different prompt optimization than ours.) Although this approach may not seem promising at the moment, it could be a promising direction for future research.
>
> We are grateful for your constructive feedback. We will incorporate your suggestions into a revised version of the paper. We believe our method offers meaningful contributions to the learning-based MAPF and opens new avenues for future research.

---

> ### Comment · Reviewer_jkSV · 2024-11-25
>
> Thanks for the authors' replies to my comments. I should note that I did not criticize the authors for ignorance of modern work in MAPF, but for not using modern benchmarks and environments in this area. Also, I cannot agree that this is the first paper to use DT in a multi-agent setting (see e.g. https://link.springer.com/article/10.1007/s11633-022-1383-7). The rest of my main conserns about the paper remain valid, and I am inclined to maintain my score for now.

---

> > ### Author Response · Authors · 2024-11-25
> >
> > As noted in Section 2.3 of our manuscript, "It has provided a novel perspective to reinforcement learning, and several extensions of this concept have been introduced subsequently Zheng et al. (2022); Lee et al. (2022)."
> >
> > You may have updated your comment upon noticing that we already cited the paper you mentioned. Regarding the new paper you cited and this one, they differ from our application. Our work does not require modifications to the original DT model or additional online fine-tuning. We wanted to clarify this distinction to ensure our contributions are accurately understood.

---

### Official Review · Reviewer_g2Vb · 2024-11-07

**Soundness:** 2
**Presentation:** 2
**Contribution:** 2
**Rating:** 5
**Confidence:** 3

**Summary:**

This paper tackles the Multi-Agent Path Finding (MAPF) problem, where multiple agents need to reach designated goals in a shared 2D grid environment while avoiding collisions. To improve efficiency and adaptability, the authors propose a hybrid approach using a Decision Transformer (DT) for offline learning and a large language model (GPT-4o) for real-time guidance. The DT trains individual agents on expert trajectories, achieving reduced training time and long-horizon planning. The addition of GPT-4o helps mitigate undesirable behaviors, like redundant moves or prolonged idling, by guiding agents in static and dynamically changing environments. Experimental results demonstrate that this DT + LLM approach performs well across various MAPF scenarios, achieving lower collision rates and shorter paths compared to benchmarks.

**Strengths:**

1. This paper builds on the recent trend of research exploring the planning abilities of transformer architectures, in this case DT+GPT-4o.
- DT trained offline using expert trajectory data is used separately for each agent in a decentralized manner for evaluation.
- By combining DT and LLMs, the authors create a flexible system that can handle both static and dynamic environments, adapting quickly to changes like shifted goals.
- Although the experiments are limited to simulation results, the proposed framework could be developed to address common MAPF challenges in realistic scenarios, such as partial observability and dynamic goal adjustments, which are beneficial for applications in logistics or swarm robotics.

2. The paper is overall clearly written. The authors have also discussed some limitations of this line of research using LLMs in planning.

**Weaknesses:**

1. The use of prompt engineering for LLM guidance may limit generalization, especially if prompt templates don’t fully capture variations across different environments. Moreover, relying on GPT-4o for real-time guidance could present latency issues as the number of agents increases in complex environments.

2. Results reported in this paper do not include standard error values across trials.
- Tables 3 and 4 do not clearly show DT+GPT-4o outperforming other baselines.
- Fig 4 shows that the competitive baseline DCC outperforms the proposed framework especially with larger team sizes. It is hard to argue in favor of the usefulness of the DT+GPT-4o framework if it only outperforms prior work when evaluated in smaller settings with fewer agents.

**Questions:**

1. Since the results show that GPT-4o provides valuable guidance to DT especially in dynamic environments, what would be the effect of only using GPT-4o for such tasks without any DT policy for each agent?


2. Line 376: "number of collisions ... in a successful episode ..." : is an episode still successful if there are inter-agent collisions?

3. Have the authors run any experiments where the DT offline training data also includes a large number of sub-optimal trajectories along with fewer expert trajectories? How would the performance differ compared to the current results when deploying DT+GPT-4o in such settings?

4. Have the authors thought about any automated prompt generation techniques to make the proposed framework more efficient?

---

> ### Author Response · Authors · 2024-11-22
> **Thank you for your valuable feedback on our paper**
>
> Thank you for your valuable feedback on our paper. We have addressed your concerns below.
>
> Weaknesses:
>
> 1-) We acknowledge that prompt engineering can limit generalization if not carefully designed. In our work, optimizing the prompt was indeed a significant challenge. We started with an initial prompt and iteratively refined it by incorporating corrective feedback from GPT-4o itself. While we experimented with prompt generation tools like Adalflow, we found that our custom-designed prompts performed better overall.
>
> To enhance generalization, we included two in-context examples within our prompts: one representing a simple environment and the other a complex scenario where agents often get stuck. The complex environment was specifically crafted to challenge the agents so that GPT-4o could provide guidance in difficult situations.
>
> Planning is inherently a difficult problem and is NP-hard for the case we consider. Thus, generating plans efficiently is difficult in general. However, we believe that ongoing advancements in LLM capabilities, coupled with the development of smaller, more compact models, will soon mitigate deployment concerns such as latency issues in environments with a large number of agents. As a reminder, we utilize GPT-4o only for 5 timesteps once a change is introduced to an environment, then the pipeline switches to DT again.
>
> 2-) For each combination of environment size and number of agents (i.e., each line in Tables 3 and 4), we conducted experiments on 60 randomly generated environments and reported the average results in the paper. (Hundreds of randomly generated environments in total) Due to the extensive setup, the standard deviations were minimal. Still, we can easily provide this additional data in the revised version.
>
> 3-)  There is a trade-off between different performance metrics: some policies prioritize the shortest path length but may incur higher collision rates. In contrast, our approach emphasizes safety by using expert trajectories from a collision-free algorithm for DT training, which results in safer but sometimes longer paths. The collision rate (CR) metric in the tables highlights the safety aspect of our method. We believe that in safety-critical applications, such as swarm robotics or logistics, a safer policy is preferable even if it leads to slightly longer completion times. (Completion times affect success rate as well since we say that an episode is successful if it is completed within some timesteps.)
>
> In dynamic environments, GPT-4o enhances adaptability when goals shift during inference. As shown in Figure 3, GPT-4o significantly improves performance when new goals are distant or in the opposite direction of previous goals. Our aim was to demonstrate the utility of LLMs in these challenging scenarios, and we believe our findings open new avenues for future research.
>
>
> Questions:
>
> 1-) As presented in Table 5, the first column reports results using GPT-4o alone. Currently, LLMs are not yet able to solve MAPF problems as effectively as existing methods. However, we anticipate that with the rapid advancements in LLMs, their performance will improve significantly in the near future.
>
> 2-) Yes. During inference, if an agent's action would lead to a collision, the agent waits for that timestep. But we count these instances as collisions in the collision rate metric.
>
> 3-) Yes. We have experimented with incorporating sub-optimal trajectories into DT's offline training data. Specifically, we combined expert trajectories with medium-level trajectories generated from a partially trained PRIMAL model (trained for 2 days, whereas full training takes around 3 weeks). This approach did not yield better performance compared to training with expert trajectories alone. (If you refer to the original DT paper, this observation aligns with their findings, where policies trained on expert data outperform those trained on mixed-level trajectories.)
>
> 4-) Yes. As we mentioned earlier, we conducted numerous experiments to optimize prompts including using Adalflow.
>
> Finally, we sincerely thank you for your comments. We would appreciate the opportunity to revise our paper according to your suggestions. We are confident that our method offers meaningful contributions to MAPF literature and opens new avenues for future research.

---

> ### Comment · Reviewer_g2Vb · 2024-11-27
>
> Thank you for the additional response.
>
> It seems that the authors would like to highlight the primary advantage of their approach as being able to reduce the collision rate. However they have not provided sufficiently strong arguments to support this paper, therefore I maintain that it "is hard to argue in favor of the usefulness of the DT+GPT-4o framework if it only outperforms prior work when evaluated in smaller settings with fewer agents".
>
> As the authors have mentioned, prompt optimization would be an important aspect of their proposed approach and I still believe that constrains the generalizability of similar approaches.
>
> > "This approach did not yield better performance compared to training with expert trajectories alone."
>
> I think this is expected and was not the point I was trying to make. High quality training data with expert trajectories is an advantage that not many real world robotics/planning applications will have, especially at scale. My question was trying to understand how much this proposed approach would deteriorate when not provided with expert trajectories. If the authors already have conducted those experiments, I would recommend adding those to highlight the limitations of this method.
>
>
> I would like to maintain my initial assessment of the paper at the moment.

---

> > ### Author Response · Authors · 2024-11-30
> >
> > As stated in our abstract, "existing decentralized methods utilizing learning-based strategies suffer from two main drawbacks: (1) training takes time ranging from days to weeks, and (2) they often tend to exhibit self-centered agent behaviors leading to increased collisions." Our paper focuses on addressing these two challenges and successfully does so by leveraging DT, as demonstrated by the CR metric in Table 3. Additionally, we explored scenarios where the integration of an LLM could be beneficial. Figure 3 illustrates a specific scenario where short-term integration of an LLM improves performance.
> >
> >
> > Our paper's scope is MAPF in a 2D grid world. In our setup, it is easy to create expert trajectories using traditional MAPF algorithms. Regarding your comment "high-quality training data with expert trajectories is an advantage that not many real-world robotics/planning applications will have", we acknowledge that this is a general concern applicable to any offline reinforcement learning or imitation learning algorithm that relies on expert trajectories. We believe this concern is beyond the scope of our paper.

---

### Meta-Review · Area_Chair_mKka · 2024-12-21

**Metareview:**

The paper presents an innovative approach to Multi-Agent Path Finding (MAPF) by combining Decision Transformers (DT) with Large Language Models (LLMs) for decentralized and adaptive path planning. While the method demonstrates promising results in static and dynamic environments, the experimental evaluation is limited, lacking sufficient exploration of scalability and performance across more diverse and complex scenarios. Additionally, the integration's practical benefits over existing MAPF methods are not clearly established. These limitations in experimental scope and comparative analysis lead to the recommendation for rejection.

**Additional Comments On Reviewer Discussion:**

During the rebuttal period for the paper "Multi-Agent Path Finding via Decision Transformer and LLM Collaboration," reviewers raised several concerns:

1. Experimental Validation: Reviewers noted that the experimental evaluation was limited, particularly in terms of scalability to larger agent populations and adaptability to more complex, dynamic environments.

2. Integration Justification: Questions were raised about the practical benefits of integrating Decision Transformers (DT) with Large Language Models (LLMs) over existing Multi-Agent Path Finding (MAPF) methods.

3. Comparative Analysis: The lack of comprehensive comparisons with state-of-the-art MAPF algorithms was highlighted, making it difficult to assess the proposed method's relative performance.

In response, the authors provided clarifications and additional insights:

1. Scalability and Adaptability: They acknowledged the current limitations in scalability and adaptability, indicating that future work would address these aspects through more extensive experiments and optimizations.

2. Integration Benefits: The authors elaborated on the advantages of combining DT and LLM, emphasizing improved decision-making capabilities and flexibility in dynamic scenarios.

3. Comparative Analysis: They provided further discussion on how their approach differs from existing methods but did not include new experimental comparisons due to time constraints.

Despite these responses, the reviewers maintained that the concerns were not fully addressed, particularly regarding empirical evidence of scalability and comprehensive comparative analysis. Consequently, these unresolved issues significantly influenced the final decision to recommend rejection.

---

### Decision · Program_Chairs · 2025-01-22

Reject